# Involvement of Spermidine in the Reduced Lifespan of *Caenorhabditis elegans* During Vitamin B_12_ Deficiency

**DOI:** 10.3390/metabo9090192

**Published:** 2019-09-19

**Authors:** Tomohiro Bito, Naho Okamoto, Kenji Otsuka, Yukinori Yabuta, Jiro Arima, Tsuyoshi Kawano, Fumio Watanabe

**Affiliations:** 1Department of Agricultural, Life and Environmental Sciences, Faculty of Agriculture, Tottori University, Tottori 680-8553, Japan; 2The United Graduate School of Agricultural Sciences, Tottori University, Tottori 680-8553, Japan

**Keywords:** arginase, *Caenorhabditis elegans*, ornithine, spermidine, vitamin B_12_

## Abstract

Vitamin B_12_ deficiency leads to various symptoms such as neuropathy, growth retardation, and infertility. Vitamin B_12_ functions as a coenzyme for two enzymes involved in amino acid metabolisms. However, there is limited information available on whether amino acid disorders caused by vitamin B_12_ deficiency induce such symptoms. First, free amino acid levels were determined in vitamin B_12_-deficient *Caenorhabditis elegans* to clarify the mechanisms underlying the symptoms caused by vitamin B_12_ deficiency. Various amino acids (valine, leucine, isoleucine, methionine, and cystathionine, among others) metabolized by vitamin B_12_-dependent enzymes were found to be significantly changed during conditions of B_12_ deficiency, which indirectly affected certain amino acids metabolized by vitamin B_12_-independent enzymes. For example, ornithine was significantly increased during vitamin B_12_ deficiency, which also significantly increased arginase activity. The accumulation of ornithine during vitamin B_12_ deficiency constitutes the first report. In addition, the biosynthesis of spermidine from ornithine was significantly decreased during vitamin B_12_ deficiency, likely due to the reduction of *S*-adenosylmethionine as a substrate for *S*-adenosylmethionine decarboxylase, which catalyzes the formation of spermidine. Moreover, vitamin B_12_ deficiency also demonstrated a significant reduction in worm lifespan, which was partially recovered by the addition of spermidine. Collectively, our findings suggest that decreased spermidine is one factor responsible for reduced lifespan in vitamin B_12_-deficient worms.

## 1. Introduction

Vitamin B_12_ (B_12_), commonly known as the red-colored vitamin, has the largest molecular mass and the most complex structure of all vitamins. B_12_ is only synthesized by certain bacteria and is primarily concentrated in higher predatory organisms in the food chain. Thus, animal-derived foods such as meat, milk, and fish are good sources of B_12_ [1]. Plant-derived foods such as vegetables and fruits contain no or trace amounts of B_12_ because plants do not require B_12_ for growth. Therefore, strict vegetarians are at a greater risk of developing B_12_ deficiency than nonvegetarians [2]. People with atrophic gastritis who develop low stomach acid output easily present the food protein–bound B_12_ malabsorption, which prevails in elderly people. Thus, strict vegetarians and elderly people are at an increased risk of developing B_12_ deficiency. In case of hereditary B_12_ deficiency, the patients defect transport proteins (intrinsic factor, a transcobalamin II, and R-binder) and the factors regarding the intracellular processing of B_12_ participating in absorption, blood transport, and intracellular metabolism of B_12_ [3].

B_12_ is converted into two coenzyme forms in living cells, namely, 5′-deoxyadenosylcobalamin and methylcobalamin, which function as coenzymes for methylmalonyl-CoA mutase (MCM, EC 5.4.99.2) [4] and methionine synthase (MS, EC 2.1.1.13) [5], respectively. MCM catalyzes the conversion of *R*-methylmalonyl-CoA to succinyl-CoA via the tricarboxylic acid (TCA) cycle in the catabolic pathways of branched-chain amino acids (valine, leucine, and isoleucine) and methionine. B_12_ deficiency leads to the excess accumulation of methylmalonic acid (MMA), which potently inhibits succinate dehydrogenase in the TCA and further leads to the disruption of normal glucose and glutamic acid metabolism [6]. MS is involved in the synthesis of methionine from homocysteine and N^5′^-methyltetrahydrofolate, and the formed methionine is converted to *S*-adenosylmethionine (SAM), which is involved in cellular methylation reactions [7]. In addition, B_12_ deficiency reportedly results in significant increases in homocysteine, which induces oxidative damage in various cellular components. Accordingly, B_12_ deficiency can induce various diseases, such as megaloblastic anemia, developmental disorders, growth retardation, and neuropathy. However, the mechanisms involved remain poorly understood.

Results from our previous study indicated that *Caenorhabditis elegans* developed severe B_12_ deficiency associated with infertility, growth retardation, and reduced lifespan when grown under B_12_-deficient conditions [8], suggesting that *C. elegans* may be a suitable model for understanding the physiological function of B_12_ and the mechanisms underlying the symptoms caused by B_12_ deficiency.

As described above, both B_12_-dependent MCM and MS function in the cellular metabolism of various amino acids. Indeed, B_12_ deficiency reportedly induces disorders of various amino acids metabolized in the B_12_-dependent pathway and indirectly affects amino acids metabolized in B_12_-independent pathways [9]. However, there is limited information available regarding whether such amino acid disorders induce these symptoms of B_12_ deficiency.

In this study, we determined free amino acid levels in B_12_-deficient *C. elegans* for clarifying the mechanisms underlying the symptoms caused by B_12_ deficiency. Therefore, ornithine was significantly increased in *C. elegans* during B_12_ deficiency. Ornithine, a precursor compound of polyamines, is formed from arginine as an essential amino acid in worms [10]. While this process is catalyzed by arginase, *C. elegans* do not contain an intact urea cycle because no homologous genes involved in the urea cycle have been found (wormbase, wormbase.org, KEGG, and https://www.genome.jp/kegg/). However, in this study, we demonstrated that B_12_ deficiency significantly increased arginase activity, leading to increased ornithine. Furthermore, the synthesis of spermidine from ornithine was significantly decreased during B_12_ deficiency, which appears to be an important factor responsible for the reduced lifespan of B_12_-deficient worms.

## 2. Materials and Methods 

### 2.1. Organisms

The N2 Bristol wild-type *C. elegans* strain was maintained at 20 °C on nematode growth medium plates using the *Escherichia coli* OP50 strain as a food source [11]. B_12_-supplemented (control) and B_12_-deficient worms were prepared as described previously [8]. B_12_-deficient worms were then transferred to B_12_-supplemented medium for three generations and used as the recovery worms [8]. For evaluating the effects of B_12_ deficiency on polyamine levels and the lifespan of the worms, we prepared control worms are grown in ornithine-supplemented (final concentration 10 μmol/L) medium for three days and B_12_-deficient worms grown in SAM-supplemented (final concentration 1 μmol/L) medium for three days.

### 2.2. Preparation of a Homogenate of Worms

Control, B_12_-deficient, and recovery worms (0.2 g wet weight each) were homogenized in 500 μL of 100 mmol/L potassium phosphate buffer (pH 7.0) on ice using a hand homogenizer (AS ONE Corp., Osaka, Japan). The homogenates were centrifuged at 15,000× *g* for 10 min at 4 °C, and these supernatants were used as crude enzymes or crude homogenates, except where otherwise noted.

### 2.3. Determination of Free Amino Acids in Worm Bodies

The homogenates of worms as described above were heated at 80 °C for 10 min and then centrifuged at 15,000× *g* for 10 min at 4 °C, the supernatants (200 μL each) were added to 100 μL of 10% (*v*/*v*) trichloroacetic acid, mixed vigorously, and centrifuged at 10,000× *g* for 10 min at 4 °C. Each supernatant (250 μL) was diluted with an equal volume of 0.25 mol/L lithium citrate buffer (pH 2.2) (Wako Pure Chemical Industries, Osaka, Japan). The diluted solution, after being filtered with Millex^®^-LH membrane filter (Merck Millipore, Darmstadt, Germany), was analyzed using a fully-automated amino acid analyzer (JEOL JLC-500/V, Nihon Denshi, Tokyo, Japan). The injection volume was 50 μL. Figure 1 indicates the outlines of the free amino acid analysis method. Appendix A shows the chromatograms of free amino acid analysis.

### 2.4. Enzyme Assays

Ornithine decarboxylase (ODC) activity was assayed using high-performance liquid chromatography (HPLC) using a fluorescence derivatization method of putrescine (Put) [12]. In detail, worms (0.05 g) grown under control, B_12_ deficiency, recovery, ornithine-supplemented control, and SAM-supplemented B_12_ deficiency conditions were homogenized in a 40 mmol/L sodium phosphate buffer (pH 7.5) (200 μL) containing 0.1 mmol/L pyridoxal phosphate and 2 mmol/L dithiothreitol and centrifuged at 15,000× *g* for 10 min at 4 °C. The supernatant fraction was then used as the crude enzyme. The reaction mixture (250 μL) contained 50 mmol/L sodium phosphate buffer (pH 7.5), 0.15 mmol/L pyridoxal phosphate, 0.3 mmol/L L-ornithine, and 100 μL of each crude enzyme. The enzymatic reaction was started with the addition of L-ornithine, incubated for an hour at 37 °C, and terminated with the addition of 50 μL of 8% (*v*/*v*) HClO_4_. The reaction mixture was then centrifuged at 15,000× *g* for 10 min at 4 °C, and the supernatant was filtered with a Millex-LH membrane filter (Merck Millipore). An aliquot (30 μL) of the filtrate was treated with 30 μL of 50 μg/mL phthaldialdehyde regent (Sigma-Aldrich, Saint Louis, MO, USA) and 50 μL of 10 mmol/L borate buffer (pH 9.5) and then incubated for 4 min at 25 °C to form a fluorescent derivative of putrescine (Put). The prepared sample (20 μL) was then injected into a reverse-phase HPLC column (Wakosil-II 5C18HG, φ4.6 × 250 mm) that had been equilibrated with 80% methanol for 1 h at 25 °C. Put was eluted with a linear gradient of methanol 80% (*v*/*v*) to 100% (*v*/*v*) at 0–11 min, 100% (*v*/*v*) at 11–14 min, 100% (*v*/*v*) to 80% (*v*/*v*) at 14–15 min, and 80% (*v*/*v*) 15–20 min) at a flow rate of 1 mL/min at 25 °C. The fluorescent derivative was then determined with fixed excitation (340 nm) and emission (455 nm) wavelengths using a Shimadzu HPLC apparatus (SCL-10A VP system controller, DGU-20A3 degassing unit, LC-10Ai liquid chromatograph, and CTO-6A column oven) equipped with a fluorescence detector (RF-530, Shimadzu).

Arginase activity was assayed using an arginase assay kit (BioAssay Systems, CA, USA) according to the manufacturer’s instructions. The absorbance at 430 nm that was changed by the enzyme reaction for 2 h was measured using a Sunrise Rainbow RC-R microplate reader (Tecan Austria GmbH, Salzburg, Austria). Arginase activity was then calculated with the calibration curve of urea as a standard. To obtain the optimum pH of worm arginase activity, various buffers ranging from pH 4.0 to 10.0 (pH 4–6, 20 mmol/L acetate-sodium acetate buffer, pH 6–8 20 mmol/L phosphate-sodium buffer, and pH 8–10 carbonate-sodium carbonate buffer) were used. Preliminary experiments indicated that the optimum temperature and pH of worm arginase was 30 °C and pH 8.0, respectively, and MnSO_4_ was required for enzyme activity (Appendix A).

Regarding N^ω^-hydroxy-L-arginine (NOHA) as a specific arginase inhibitor, B_12_-deficient worms were grown in B_12_-deficient medium containing NOHA (final concentration at 10 nM) for 3 days before the assay of arginase activity assay. The arginase inhibitor was used at a concentration shown to not influence the growth of the worms.

### 2.5. Other Assays

SAM and *S*-adenosylhomocysteine (SAH) were assayed by HPLC according to the modified method of Wang et al. [13]. Each worm (0.05 g wet weight) was homogenized in 200 μL of 0.4 mol/L HClO_4_ on ice. After centrifugation at 15,000× *g* for 15 min at 4 °C, the supernatant was filtered through a Millex-LH membrane filter (Merck Millipore). Each prepared extract (20 μL) was injected into the Shimadzu HPLC apparatus equipped with an SPD-10A VP UV/VIS detector (Shimadzu). The analytical method was developed with a flow rate of 1 mL/min, column temperature of 35 °C, and UV detection at 254 nm. A Wakosil 5C18HG column (φ4.6 × 250 mm, Wako) was equilibrated with 80% solvent A (50 mmol/L NaH_2_PO_4_ (pH 3.0) containing 8 mmol/L octanesulfonic acid sodium salt) and 20% solvent B (methanol). SAM and SAH were eluted over a 13-min gradient as follows: 80% solvent A-20% solvent B at 0 min to 3 min, a liner gradient of 60% solvent A-40% solvent B at 3 min to 9 min, and 60% solvent A-40% solvent B at 9 min to 13 min. As modification points, column and gradient conditions were changed as described above.

Polyamines, such as Put and Spermidine (Spd), were analyzed using a standard HPLC method [14]. Worms (0.05 g wet weight) grown under control, B_12_ deficiency, recovery, ornithine-supplemented control, and SAM-supplemented B_12_ deficiency conditions were homogenized in 200 μL of 0.3 mol/L HClO_4_ using a hand homogenizer on ice. After centrifugation (15,000× *g*, 10 min at 4 °C), the supernatants were used as extracts. Polyamines were derivatized with benzoyl chloride, and the reaction mixture (1110 μL) was mixed with 1 mL of 2 mol/L NaOH, 10 μL of benzoyl chloride, and 100 μL of extract and incubated for 40 min at room temperature (25 °C). The reaction was terminated with the addition of 2 mL of a saturated NaCl solution, and benzoylated polyamines were extracted with 2 mL of cold diethyl ether. After centrifugation (1500× *g*, 5 min at 4 °C), an aliquot (1 mL) of diethyl ether was recovered, evaporated to dryness under a nitrogen flow, and dissolved in 200 μL of 50% (*v*/*v*) methanol. Authentic Put, Spd, and 1,6-diaminohexane were treated under the same conditions, and 1,6-diaminohexane was used as the internal standard. Each prepared extract was then filtered through a Millex-LH membrane filter (Merck Millipore) and injected into the Shimadzu HPLC apparatus equipped with a UV/VIS detector (Shimadzu). The analytical method was developed with a flow rate of 1 mL/min, column temperature of 35 °C, and UV detection at 254 nm. A Wakosil-II 5C18HG column (φ4.6 × 250 mm, Wako) was equilibrated with 50% methanol. Polyamines were eluted with a linear gradient of methanol 50% (*v*/*v*) to 55% (*v*/*v*) at 0–5 min, 55% (*v/v*) to 60% (*v*/*v*) at 5–20 min, and 60% (*v*/*v*) to 80% (*v*/*v*) at 20–25 min. Figure 2 shows the scheme of derivatization reaction of polyamines. Appendix A showed the chromatograms of polyamine analysis.

### 2.6. Quantitative PCR Analysis

Control worms were grown in ornithine (final concentration 10 μM)-supplemented control medium for three days to evaluate the effects of accumulated ornithine on worm polyamine metabolism. B_12_-deficient worms were grown in SAM (final concentration 1 μM)-supplemented B_12_-deficient medium for three days to clarify the effects of added SAM on worm polyamine metabolism. Control worms, ornithine-supplemented control worms, B_12_-deficient worms, and SAM-supplemented B_12_-deficient worms were used as samples in the following quantitative PCR analysis.

Total RNA was prepared from the worms using Sephasol^®^-RNA1 (Nacalai Tesque, Kyoto, Japan). Total RNA was used to synthesize cDNA using the PrimeScript™ RT Reagent Kit with gDNA Eraser (Takara Bio, Shiga, Japan). Primer sets were designed using GENETYX software (GENETYX Corporation, Tokyo, Japan) as follows: ornithine decarboxylase (*odc-1*), forward primer sequence, 5′-AAGGGCTCGGATTCAAGATGGA-3′, and reverse primer sequence, 5′-TTCAGCAATCAGGCGCTTGT-3′, *S*-adenosylmethionine decarboxylase (*smd-1*), forward primer sequence, 5′-ATGCGAGCCGGTATTGACAAG-3′, and reverse primer sequence, 5′-GGATGGTTGCGTATTGGTCAGT-3′, spermidine synthase (*spds-1*), forward primer sequence, 5′-AACGGGATGAGTTCTCCTACCA-3′, and reverse primer sequence, 5′-ACTCGTGCTTCAAGACCTCTCT-3′, human arginase type 1 erythroid. T21F4.1 was designed as follows: T21F4.1, forward primer sequence, 5′-CACGTGGGTGAGATAATATGCC-3′, and reverse primer sequence, 5′-CGTCGTCCGATACATTTCCTTC-3′, β-actin (*act-1* as an internal standard), forward primer sequence, 5′-TCCAAGAGAGGTATCCTTACCC-3′, and reverse primer sequence, 5′-CTCCATATCATCCCAGTTGGTG-3′. A CFX Connect^TM^ Real-Time System (Bio-Rad Laboratories, Inc. Hercules) with SYBR Premix Ex Taq (Takara Bio) was used to perform qPCR.

### 2.7. Lifespan Analysis

Control and B_12_-deficient worms were grown in ornithine (final concentration 10 μM)-, SAM (final concentration 1 μM)-, or Spd (final concentration 10 μM)-supplemented (final concentration 10 μM) medium for three days to evaluate the effects of ornithine, SAM, and Spd on the lifespan of *C. elegans*. Control worms, ornithine-supplemented control worms, Spd-supplemented control worms, B_12_-deficient worms, and Spd-supplemented B_12_-deficient worms were used as samples in this experiment. The lifespan of the worms was measured by the method of Apfeld and Kenyon [15]. Each worm was picked (approximately 10 animals per plate) and allowed to grow at 20 °C until they laid eggs. The next-generation worms (2- or 3-day-old) were picked to plates containing 5-fluoro-2′-deoxyuridine, which inhibits the worms from laying eggs. Worms were then tapped every 24 h and scored as dead when they did not move with repeated taps. The average lifespan of each worm was defined as a 50% survival rate obtained from the survival curves of each worm.

### 2.8. Protein Quantitation

Protein quantification was determined by the Bradford method [16], with ovalbumin as the standard.

### 2.9. Statistical Analysis

All data were evaluated by one-way ANOVA (non-parametric test), and a post-hoc analysis was performed using Tukey’s multiple comparison tests. Analyses were performed with GraphPad Prism 3 for Windows version 2.01 (GraphPad Software Inc., La Jolla, CA, USA). All data are presented as the mean ± SEM. Differences were considered statistically significant at *p* < 0.05.

## 3. Results

### 3.1. Effect of B_12_ Deficiency on Free Amino Acids in C. elegans

Table 1 shows the effect of B_12_ deficiency on free amino acids in *C. elegans*. Various amino acids (such as valine, leucine, and isoleucine) metabolized as part of the TCA cycle via the B_12_-dependent MCM pathway were increased significantly during B_12_ deficiency because MCM activity was significantly decreased during B_12_ deficiency [9]. Decreased MS activity induced a significant decrease in methionine, with a concomitant significant increase in cystathionine. In addition, threonine, lysine, aminoadipic acid, and ornithine levels were significantly increased during B_12_ deficiency. Significantly changed levels of such amino acids were mostly recovered to control levels when B_12_-deficient worms were grown for three generations under B_12_-supplemented conditions (recovery). These results indicate that B_12_ deficiency induces metabolic disorders of various amino acids in *C. elegans*. However, the accumulation of ornithine during B_12_ deficiency, to the best of our knowledge, is the first report.

### 3.2. Effects of B_12_ Deficiency on Arginase Activity and the Levels of mRNAs Encoding Putative Arginase in C. elegans

Ornithine is formed from arginine by the action of arginase involved in the urea cycle. However, *C. elegans* do not have an intact urea cycle. When we checked into the wormbase database, T21F4.1 [17] was registered as the orthologous gene human arginase type 1 and 2 erythroid variants in the wormbase database. Sequence analysis showed that *C. elegans* T21F4.1 has 27–35% and 68–73% of sequence identity and similarity, respectively, with *homo sapiens*, *Bacillus caldovelox*, *Bacillus subtilis*, and *Rattus norvegicus* arginases (Figure 3).

To investigate the cause of ornithine accumulation caused by B_12_ deficiency, arginase activity was assayed. Worm arginase activity increased by approximately 1.7-fold in the B_12_-deficient worms compared with the control worms (Figure 4a). The increased arginase activity in B_12_-deficient worms was recovered to control levels when the B_12_-deficient worms were grown under-recovery and NOHA-treated (a potent inhibitor of arginase) conditions. These results suggested that ornithine accumulated by the increased arginase activity observed during B_12_ deficiency.

The mRNA expression level of T21F4.1 was noted to exhibit the tendency to increase in the B_12_-deficient worms (Figure 4c). However, this increased mRNA expression was not found to be significant. Nonetheless, T21F4.1 mRNA expression in B_12_-deficient worms was recovered to control levels when B_12_-deficient worms were grown under-recovery conditions. When B_12_-deficient worms were treated with NOHA, the addition of NOHA did not affect the mRNA expression level of T21F4.1, but did significantly decrease arginase activity, leading to decreased worm ornithine contents (Figure 4a,b). These results suggested that B_12_ deficiency significantly increased arginase activity, which may be attributable to the mRNA expression of T21F4.1, leading to increased ornithine.

### 3.3. Effect of B_12_ Deficiency on Polyamine Levels of C. elegans

B_12_ deficiency showed that worm SAH increased significantly with a decrease in SAM (Figure 5a,b). The SAM/SAH ratio as an index of the cellular methylation reaction was decreased to approximately 39% of the control levels (Figure 5c). The decreased SAM/SAH ratio was completely recovered to control levels when B_12_-deficient worms were grown for three generations under B_12_-supplemented conditions. These results indicate that B_12_ deficiency leads to a disorder of cellular methylation reactions in *C. elegans* as well as in mammals.

As shown in Figure 5a,b, B_12_ deficiency did not affect cellular Put but did significantly decrease Spd, which was approximately 83% of that in control worms. To clarify why Spd significantly decreased in *C. elegans* during B_12_ deficiency, the mRNA expression of enzymes involved in the biosynthesis of polyamines from ornithine was determined. Remarkably, the enzyme activity and mRNA expression levels of ODC (Figure 6c,d) were increased in *C. elegans* during B_12_ deficiency. This did not affect the mRNA levels of Spd synthase (SPDS, *spds-1*), but did lead to significantly decreased SAM decarboxylase (SAMDC) (*smd-1*) mRNA levels (Figure 6e,f). These altered mRNA levels were recovered to control levels when B_12_-deficient worms were grown for three generations under B_12_-supplemented conditions.

The addition of ornithine (10 μM) to control worms significantly increased ornithine levels up to approximately 0.58 μmol/g of worm, at which the ornithine levels were considered greater than those of B_12_-deficient worms. The added ornithine did not affect levels of Spd content, ODC activity, or *odc-1* and *smd-1* mRNA expression levels (Figure 6b–e). However, these levels that were changed during B_12_ deficiency were mostly recovered to control levels in SAM (1 μM)-treated B_12_-deficient worms (Figure 6b–e). These results suggested that the decreased Spd in B_12_-deficient worms was due to the reduction of SAM as a substrate for SAMDC.

### 3.4. Effect of Supplementation of Spd on the Lifespan of C. elegans

The maximal lifespan of B_12_-deficient worms was reduced to 22 days, compared with a lifespan of 28 days in control worms (Figure 7). The average lifespan was decreased to a greater extent in B_12_-deficient worms (15.48 days) than in control worms (20.84 days). When B_12_-deficient worms were treated with Spd, the average lifespan (15.48 days) was extended to 17.42 days (an increase up to 1.94 days) (Figure 7). In the case of control worms, the average lifespan of Spd-treated worms was 21.58 days (an increase of only 0.74 days). The addition of Spd to B_12_-deficient worms showed a significantly extended average lifespan, compared with that of B_12_-deficient worms. However, no significant difference was noted between the average lifespan of B_12_-deficient worms and Spd-treated B_12_-deficient worms, considering the extended average lifespan of Spd-supplemented control worms. These results indicated that the reduced lifespan of B_12_-deficient worms was recovered by approximately 22.4% through the addition of Spd, suggesting that the reduced lifespan of worms during B_12_ deficiency is partly caused by decreased Spd.

## 4. Discussion

B_12_ deficiency reportedly disrupts the TCA cycle due to the decreased activity of succinate dehydrogenase [6], which is inhibited by MMA accumulation observed during B_12_ deficiency. This process then leads to significant increases in lysine and 2-aminoadipaic acid levels, with decreased formation of aspartic acid from oxaloacetic acid and glutamic acid through the actions of aspartate aminotransferase. Similarly, an increase was observed in the excretion of threonine, serine, valine, isoleucine, and lysine in urine in B_12_-deficient rats [5]. In addition, the mRNA and protein expression levels of hepatic serine dehydrogenase (a B_12_-independent enzyme), which catalyzes the conversion of serine and threonine to pyruvate and 2-oxobutyrate, respectively, were significantly lowered in B_12_-deficient rats [9]. However, the serine level was not found to be increased in B_12_-deficient worms.

Furthermore, we demonstrated for the first time, to the best of our knowledge, that B_12_ deficiency significantly increased the ornithine level, which is involved in polyamine biosynthesis [12]. The *E. coli* OP50 strain as the worm diet did not contain ornithine, and other amino acids showed no changes in *E. coli* cells grown in both B_12_-supplemented and B_12_-deficient conditions (data not shown). Ornithine is usually formed from arginine through the actions of arginase (EC 3.5.3.1), which is involved in the mammalian urea cycle [18]. However, *C. elegans* do not have an intact urea cycle because no homologous gene encoding enzymes involved in the urea cycle has been found in *C. elegans* (wormbase, wormbase.org, KEGG, www.genome.jp/kegg). Thus, *C. elegans* absolutely require arginine as an essential amino acid, as well as lysine, threonine, isoleucine, leucine, valine, methionine, phenylalanine, tryptophan, and histidine [10]. In particular, arginine reportedly functions as an energy-saving compound in *C. elegans* after being converted into arginine phosphate [10]. Arginine is not only metabolized by arginase to form ornithine and urea, but it is also metabolized by nitric oxide synthase (NOS) (EC 1. 14.13.39) to form nitric oxide (NO) and citrulline [19]. In this study, B_12_ deficiency did not affect arginine levels in *C. elegans* (Table 1), possibly to be able to sufficiently supply arginine from the diet. Although B_12_ deficiency reportedly increased cellular NO levels in *C. elegans* [20], a full complement of homologs of NOS has not yet been identified in the genome of *C. elegans*, suggesting that NO produced by bacteria (in the diet) in the worm gut may diffuse into the tissues. Our previous study [20] indicated that B_12_-deficiency in worms induces severe oxidative stress, which likely stimulates arginase activity because oxidative species reportedly increase arginase activity in bovine aortic endothelial cells [21].

Remarkably, urea, which is formed from arginine by arginase, has a tendency to increase in B_12_-deficient *C. elegans* (Table 1). Similarly, urea has been reportedly excreted in *C. elegans* [22] and the other free-living nematode, *Panagrellus redivivus* [23], implying the occurrence of arginase in *C. elegans*. These observations suggest that ornithine is enzymatically synthesized by arginase from arginine in *C. elegans* as an essential amino acid. The amino acid disorders observed in *C. elegans* during B_12_ deficiency are summarized in Figure 8.

For measuring the arginase activity in *C. elegans*, some enzymatic properties of arginase were determined using a homogenate of B_12_-deficient worms (Appendix A). Most arginases have the highest activity in alkaline pH ranges and require Mn(II) ion or Co(II) ion as a cofactor [24]. Similarly, arginase in the worms showed the highest activity at pH 8.0 and absolutely required Mn(II) ion. Even though arginase activity (approximately 0.4–0.8 pmol/min/mg protein) was detected in a homogenate of *C. elegans*, worm enzyme activity was significantly lower than that of human liver (5 μmol /min/mg protein) and red blood cells (0.05 μmol /min/mg protein) [25]. These results and perspectives indicated that *C. elegans* has the arginase with weak function. However, the gene product of T21F4.1 could not be characterized as arginase in detail because of the unsuccessful cloning and overexpression of the T21F4.1 gene. Peroxisome proliferator activated-receptor (PPAR) α is considered to possess the regulation function of the expression of genes involved in β-oxidation in fatty acids. Ahmad et al. [26] found that the protein expression of arginase 1 increases through the PPAR signaling pathway in B_12_-deficient rats. The mRNA level of T21F4.1 may increase due to the activation of PPAR signaling pathway by lipid metabolic disorder caused by B_12_ deficiency. When B_12_-deficient worms were grown in NOHA-treated conditions, ornithine was decreased to control level with a decrease in the arginase activity (Figure 4a,b). These results suggested that the accumulation of ornithine during B_12_ deficiency was a result of the increase in arginase activity. In addition, urea is well-known as an antioxidant in the body [27]. Urea formed by the action of arginase may function as an antioxidant because B_12_ deficiency induced severe oxidative stress in *C. elegans*.

As described above, the increased arginase activity in *C. elegans* during B_12_ deficiency resulted in increased ornithine, which is a precursor amino acid of polyamines. The reduced activity of B_12_-dependent MS by B_12_ deficiency [8] further decreased cellular SAM levels (Figure 5a), leading to a significant decrease of Spd (Figure 6b) in *C. elegans.* A significant reduction in the cardiac and serum SAM/SAH ratio has been reported in mammals during B_12_ deficiency [28,29] likewise B_12_-deficient worms. In addition, Spd reportedly stimulated B_12_-dependent MS activity [30], suggesting that reduced Spd may further decrease B_12_-dependent MS activity, thus increasing homocysteine [8] and leading to the further development of oxidative damage [20]. The biosynthesis of Put and Spd is highly regulated by two key enzymes, namely, ornithine decarboxylase (ODC, EC.4.1.1.17) and SAMDC (EC.4.1.1.50), are strongly regulated by feedback mechanisms at several levels, including transcriptional, translational, and post-translational [31]. In particular, ODC activity reportedly increased in murine L1210 leukemia cells [32] and in the spinal cords of totally gastrectomized rats [33] by the decrease in SAM. Spermine, which is a polyamine, was not detected in *C. elegans* in this study. *Caenorhabditis elegans* are not believed to contain spermine because no homologous gene of spermine synthase (EC. 2.5.1.22) has been found [34]. In general, *odc-1* and *smd-1* gene expressions were strictly regulated by substrates and productions [31]. From these results and suggestions, it was difficult to interpret the response of changes of odc-1 and smd-1 gene expression in the case of *C. elegans* not having spermine.

Polyamines play multiple roles in cell growth and death, including aging, neurodegenerative diseases, and metabolic disorders [35]. In particular, Spd has shown life-prolonging effects in various organisms, including in mammals and *C. elegans*. Indeed, Eisenberg et al. [36] have demonstrated that the administration of exogenous Spd significantly extends lifespan through epigenetic modifications, the inducement of autophagy, and a decline in necrosis using yeast, flies, *C. elegans*, and human immune cells as aging models. Remaining factors underlying the reduced lifespan observed during B_12_ deficiency may be due to inhibition of the TCA cycle, which would disrupt normal energy production, leading to oxidative damage in various cellular components, and reduced methylation reactions involving cellular compounds [8,20].

## 5. Conclusions

Collectively, our novel findings indicated that B_12_ deficiency induced both significant increases in ornithine and decreases in Spd, due to the reduction of SAM and/or the SAM/SAH ratio through decreased B_12_-dependent MS activity. Although such a phenomenon has not been reported in B_12_-deficient mammals, evidence exists that changes in polyamine levels are associated with aging and disease [37]. Thus, polyamines may prove to be key compounds underlying the symptoms of various health conditions (megaloblastic anemia, developmental disorders, growth retardation, and neuropathy) caused by B_12_ deficiency. To further clarify the accurate functions of polyamines associated with the symptoms of B_12_ deficiency in mammals, further studies are warranted.

## Figures and Tables

**Figure 1 metabolites-09-00192-f001:**
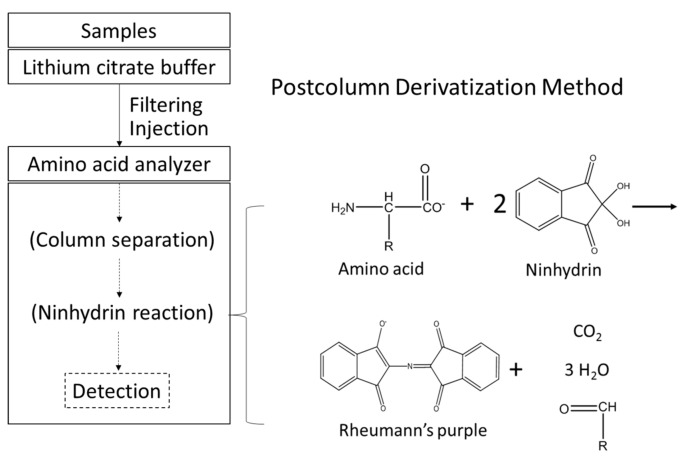
Outline of the amino acid analysis method.

**Figure 2 metabolites-09-00192-f002:**
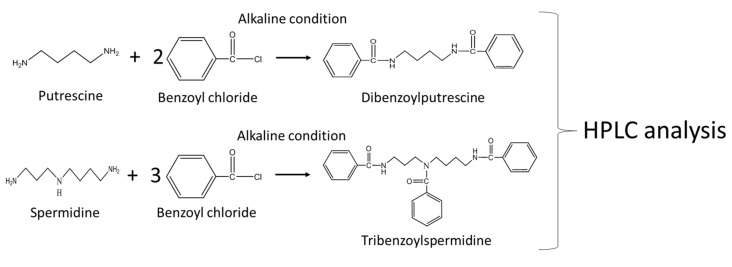
Scheme of derivatization of polyamines.

**Figure 3 metabolites-09-00192-f003:**
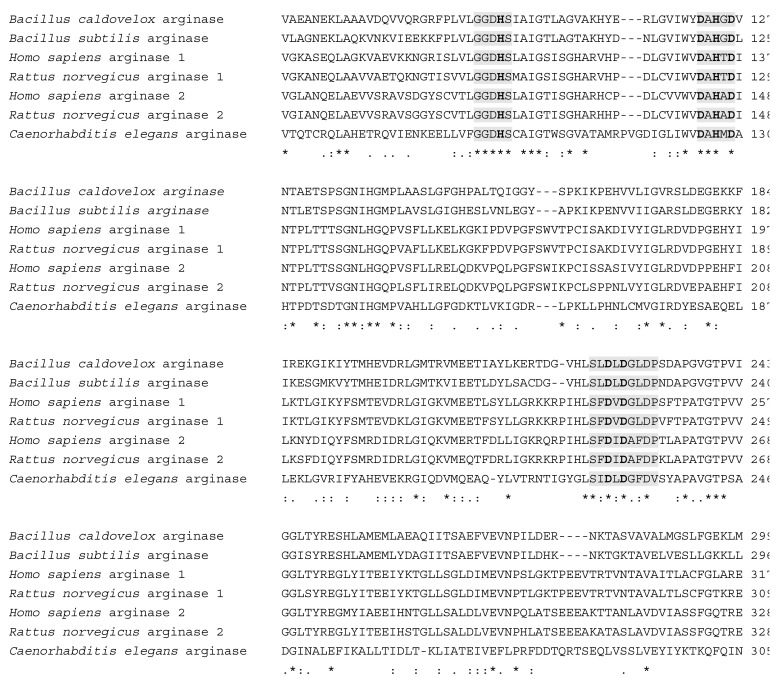
Amino acid sequence alignments of arginases and putative arginase T21F4.1 in *C. elegans* using CLUSTAL W. Alignment of *C. elegans* T21F4.1 (accession no.NP_001257027.2) with *Bacillus caldovelox* arginase (accession no. S68863), *Bacillus subtilis* arginase (accession no. NP_391912.1), *Homo sapiens* arginase 1 (accession no. NP_001231367.1), *Rattus norvegicus* arginase 1 (accession no. NP_058830.2), *Homo sapiens* arginase 2 (accession no. NP_001163.1), and *Rattus norvegicus* arginase 2 (accession no. NP_062041.1). Highlighted areas indicate the amino acid signature motifs characteristic of the arginase family. Residues in bold are critical for the formation of a bimetallic cluster at the active-site. The analysis reveals that *C. elegans* T21F4.1 has 27–35% and 68–73% of sequence identity and similarity, respectively, with other arginases. The N- and C-terminal residues are omitted for clarity.

**Figure 4 metabolites-09-00192-f004:**
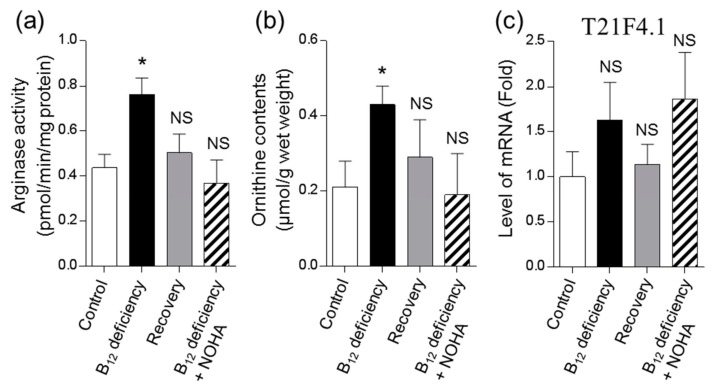
Effects of vitamin B_12_ deficiency on arginase activity and the mRNA expression of the arginase orthologous gene, T21F4.1, in *C. elegans*. Arginase activity (**a**), ornithine contents (**b**), and mRNA expression levels of arginase orthologous gene, T21F4.1, (**c**) were determined in B_12_-deficient and control worms. After B_12_-deficient worms were grown for three generations under B_12_-supplemented conditions, these values were determined in the B_12_-treated worms (shown as “Recovery”). After B_12_-deficient worms were treated with N^ω^-hydroxy-L-arginine (NOHA) as a potent inhibitor of arginase, these values were determined (shown as “B_12_ deficiency + NOHA”). Data represent the mean ± SEM of three independent experiments. * *p* < 0.05 versus Control group. NS represents no significant differences.

**Figure 5 metabolites-09-00192-f005:**
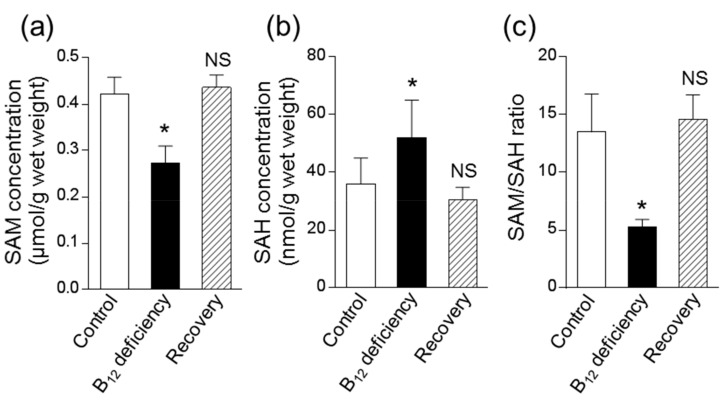
Effects of vitamin B_12_ deficiency on S-adenosylmethionine and S-adenosylhomocysteine contents of *C. elegans*. S-Adenosylmethionine (SAM, **a**) and S-adenosylhomocysteine (SAH, **b**) were determined in B_12_-deficient and control worms using HPLC, and then the SAM/SAH ratio (**c**) was calculated. After B_12_-deficient worms were grown for three generations under B_12_-supplemented conditions, these values were determined in the B_12_-treated worms (shown as “Recovery”). Data represent the mean ± SEM of three independent experiments. * *p* < 0.05 versus Control group. NS represents no significant differences.

**Figure 6 metabolites-09-00192-f006:**
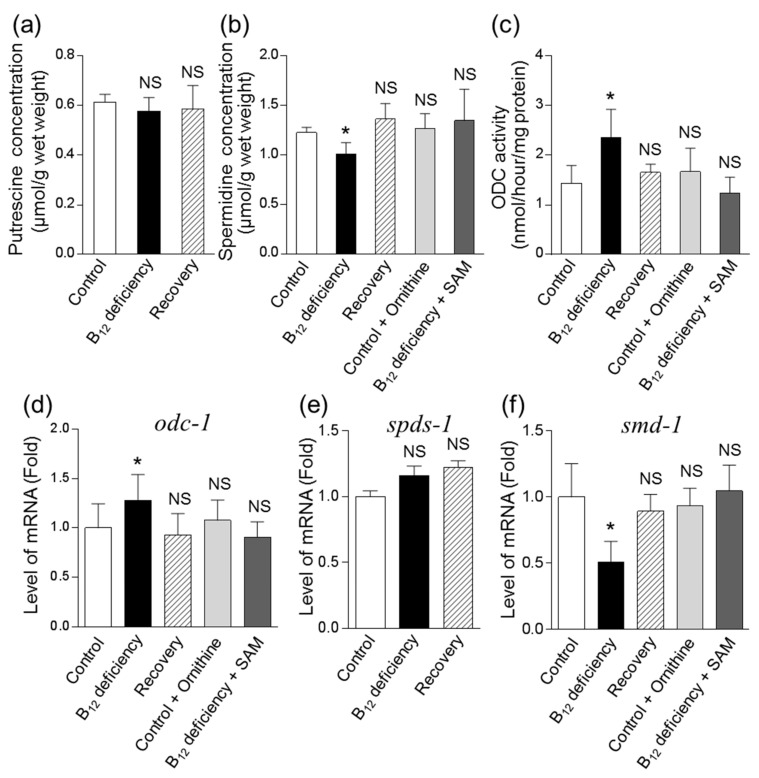
Effects of vitamin B_12_ deficiency on polyamine (putrescine and spermidine) levels and the mRNA expression levels of genes encoding enzymes involved in the biosynthetic pathway of spermidine from ornithine in *C. elegans*. Putrescine (**a**) and spermidine (**b**) levels and ornithine decarboxylase (ODC, **c**) activity were determined in control and B_12_-deficient worms using HPLC. Control worms and B_12_-deficient worms were grown in ornithine (final concentration 10 μM)-supplemented and S-adenosylmethionine (SAM) (final concentration 1 μM)-supplemented medium for three days. Ornithine-supplemented control worms and SAM-supplemented B_12_-deficient worms were also analyzed and shown as Control + Ornithine and B_12_ deficiency + SAM, respectively. The mRNA expression levels of the orthologous genes of human ODC (*odc-1*), spermidine synthase (*spds-1*), and S-adenosylmethionine decarboxylase (*smd-1*) represent (**d**), (**e**), and (**f**), respectively. Data represent the mean ± SEM of three independent experiments. * *p* < 0.05 versus Control group. NS represents no significant differences.

**Figure 7 metabolites-09-00192-f007:**
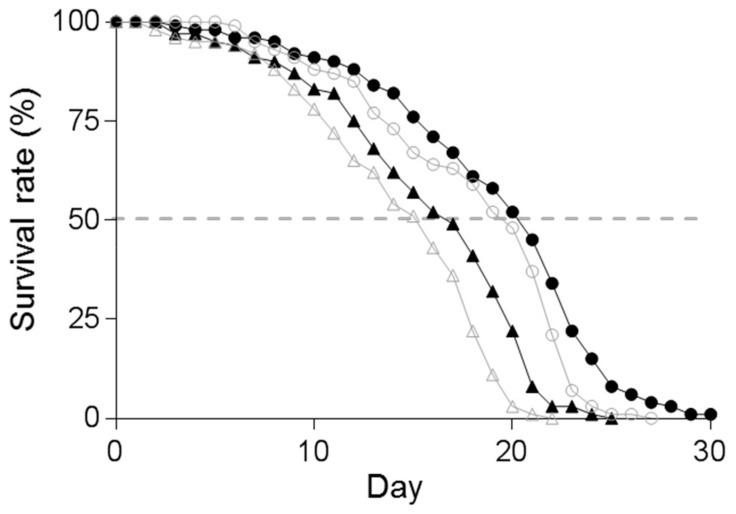
Effect of spermidine on lifespan in control and B_12_-deficient worms. Control worms (◯), B_12_-deficient worms (△), control worms grown under the supplementation of spermidine (●), and B_12_-deficient worms grown under the supplementation of spermidine (▲). The gray-colored graphs represent control and B_12_-deficient worm data in this Figure. The lifespan of worms (mean days, *n* = number of worms scored) was measured as described in the text, control, 20.84 days, *n* = 126, B_12_ deficiency, 15.48 days, *n* = 113, spermidine-supplemented control, 21.58 days, *n* = 143, spermidine-supplemented B_12_ deficiency, 17.42 days, *n* = 106.

**Figure 8 metabolites-09-00192-f008:**
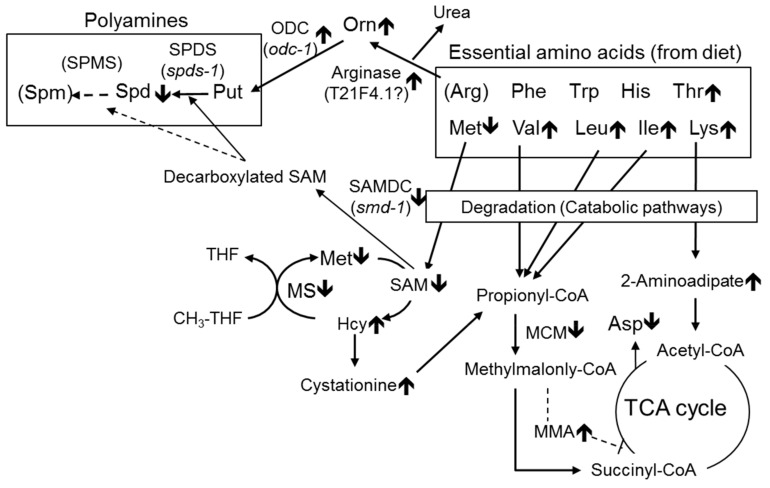
Summary of the metabolic disorders of amino acids observed in *C. elegans* during vitamin B_12_ deficiency. MCM, methylmalonyl-CoA mutase, MS, methionine synthase, MMA, methylmalonic acid, SAM, S-adenosylmethionine, THF, tetrahydrofolate, 🡱, increase, 🡳, decrease.

**Table 1 metabolites-09-00192-t001:** Concentrations of free amino acids and other compounds in *C. elegans*.

	*C. elegans* (mmol/g wet weight)		
	Control	B_12_ Deficiency	Recovery
Amino acids			
Aspatic acid	1.29 ± 0.22	0.58 ± 0.22	1.37 ± 0.29
Threonine	0.55 ± 0.04	0.80 ± 0.10 *	0.49 ± 0.08
Serine	0.76 ± 0.07	0.93 ± 0.17	0.82 ± 0.08
Asparagine	0.57 ± 0.07	0.41 ± 0.07	0.61 ± 0.05
Glutamic acid	3.27 ± 0.38	3.41 ± 0.28	3.32 ± 0.45
Glutamine	1.32 ± 0.22	1.28 ± 0.29	1.27 ± 0.15
Glycine	0.73 ± 0.06	0.78 ± 0.05	0.63 ± 0.11
Alanine	5.80 ± 0.66	6.05 ± 0.22	5.53 ± 0.39
Valine	0.58 ± 0.06	1.03 ± 0.13 *	0.52 ± 0.09
Methionine	0.19 ± 0.03	0.12 ± 0.03 *	0.22 ± 0.04
Isoleucine	0.37 ± 0.05	0.61 ± 0.08 *	0.36 ± 0.05
Leucine	0.70 ± 0.09	1.07 ± 0.12 *	0.63 ± 0.10
Tyrosine	0.22 ± 0.02	0.30 ± 0.02	0.24 ± 0.04
Phenylalanine	0.35 ± 0.05	0.47 ± 0.10	0.40 ± 0.06
Ornithine	0.21 ± 0.07	0.43 ± 0.05 *	0.29 ± 0.10
Lysine	0.43 ± 0.04	0.71 ± 0.01 *	0.40 ± 0.04
Histidine	0.34 ± 0.04	0.55 ± 0.12	0.39 ± 0.05
Arginine	0.88 ± 0.10	1.14 ± 0.16	0.81 ± 0.05
Hydroxyproline	0.07 ± 0.06	0.03 ± 0.01	0.05 ± 0.03
Others			
Cystathionine	0.57 ± 0.06	3.57 ± 0.31 *	0.78 ± 0.13
Aminoadipic acid	0.79 ± 0.09	1.44 ± 0.12 *	0.92 ± 0.15
Urea	1.66 ± 0.52	2.44 ± 0.90	1.84 ± 0.66

Results are presented as means ± SEM (*n* = 5). Values with asterisk (*) is significantly different (*p* < 0.05).

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
