# Peer review of "Involvement of Spermidine in the Reduced Lifespan of Caenorhabditis elegans During Vitamin B12 Deficiency"

_metabolites, 2019, doi:10.3390/metabo9090192_

Round 1
Reviewer 1 Report
Bito et al describe a study investigating the effect of vitamin B12 deficiency on C. elegans, the study is interesting though I am unsure if metabolites is the best fit. The manuscript would greatly benefit from a more comprehensive introduction as the current version doesn’t explain the significance of this study. The work also needs significantly more information regarding the methods being used. The discussion is well written and serves the manuscript well.
Abstract:
The abstract would benefit from a line or two on the back ground to the study, why study this in the first place?
Introduction:
The introductions dives right into describing the biochemistry associated with vitamin B12, there is not description as to why this study was required. What gap in the literature is it filling? What are the causes of vitamin B12 deficiency? Are they dietary issues or medical conditions? Is there a difference between these? Who is most affected?
2.4 Enzyme assays
Page 2 lines 79-80: What was the method used? There needs to be some description, what was the detector used? What was the gradient. If it’s a modified method from the references work then it needs much more information.
Page 2 line 80: What were these various conditions?
Page 3 lines 90-96: Were the mobile phases pure water and MeOH? No acid/base or salt? What was the gradient conditions? Injection volume? Column temperature e.t.c.
2.5 Other assays:
Page 3 lines 111-112: What was the method its stated its modified but no details are given as such the experiments cannot be repeated elsewhere.
Page 3 line 123: What various conditions?
2.9 statistical analysis:
Was the data checked for normality?
3.1 Effect of B12 deficiency on free amino acids in C.elegans
Page 5 line 190-192: Have the authors proved it’s the same in mammals? They were not assessed here so there must be a reference to back this statement up
Author Response
Please see attachfile.

Reviewer 2 Report
In this manuscript Bito et al have analysed the consequences of Vitamin B12 deficiency using the organism C. elegans as a model. In a previous article the authors reported that C. elegans is very sensitive to the lack of this vitamin, which causes infertility, growth retardation and reduced lifespan. In the present work the authors have thoroughly analysed the biochemical consequences of vitamin B12 deficiency.
The authors have measured the levels of free aminoacids. As it could be expected, they have found a decrease in Met (due to defective MS activity) and the increase in several aminoacids whose degradation through the TCA depends on MCM (Thr, Val, Leu, Ile, Lys). These alterations look quite similar to those described in mammals. Moreover, as a novelty the authors have also found an increase in the level of ornithine. Ornithine is a precursor of poliamines synthesis, and the decrease of spermidine (spd) is also observed. These changes seem to be mainly related with altered expression of some of the enzymes involved in the synthesis of these metabolites, and with the decrease of SAM.
In agreement with their previous results, the authors have found a decrease of C. elegans lifespan upon vitamin B12, which seems to be partially related with the decrease of spermidine levels, but that does not seem to be related with the increase in ornithine. These results are especially interesting since they suggest the importance of Vitamin B12 for the control of lifespan through the regulation of spermidine levels, among other factors.
Please find specific comments below.
Main comments:
The increase in ornithine levels seems to be related with the increase in arginase activity, an enzyme that has not been characterised in elegans before. In figure 1a the authors measured such activity. According to the information provided in the methods section different conditions were tested, finding as optimal conditions pH 8.0 and 30ºC. I wonder whether the optimal pH is similar to the putative orthologue enzyme in mammals. If this is so, that would somehow support the occurrence of arginase in C. elegans. In order to characterise the arginase activity elegans the authors have analysed a putative orthologue (TF21F4.1). The information provided in the text regarding this issue is very scarce. In the methods section the authors just list a couple of oligonucleotides used for TF21F4.1 detection without any further explanation. They also list a couple of oligos for human arginase type 1 erythroid; what have they been used for? This is an important point in the manuscript, so it should be explained in the text more clearly. Moreover, it might be interesting to state which is the percentage of similarity between mammalian arginase and TF21F4.1. Does TF21F4.1 shows a putative arginase catalytic domain? All in all, this would support the occurrence of an arginase in C. elegans, even more bearing in mind that the cloning of TF21F4.1 was unsuccessful. Figure 1c suggests that the increase in ornithine upon Vit. B12 deficiency might be due to the increase expression of TF21F4.1 at the mRNA level. However, the data do not reach statistical significance, probably due to the number of repeats (n=3). It might be interesting to increase the n in figure 1c to see if statistical significance can be reached. That would reinforce the results shown in this figure. If the increase in ornithine is due to the increased arginase activity, one would expect an increase in urea and a decrease in arginine. However, no changes were detected in these two metabolites. Please clarify this issue further. Moreover, since elegans does not have urea cycle, what would be the meaning of urea production in the worm. Figure 3 is missing, although the results are described in the text, it is quite difficult to asses this part of the manuscript without a figure. In figure 4a the authors show that vitamin B12 deficiency reduces elegans lifespan, which could be partially recovered by spermidine supplementation. Spermidine extend the average lifespan from 15.48 to 17.42 days (figure 4b). No statistical analyses are shown in figure 4. It might be interesting to know if the lifespan extension caused by spermidine is statistically significant. The decrease in spermidine might be mainly related with the reduction of SAM levels, and probably also with the reduction of SAMDC mRNA levels. Bearing this in mind I wonder why putrescin levels are not affected since SAM would be also required for its synthesis. The increase in ornithine does not seem to be related to either spermidine decrease or the reduced lifespan. So, what is the importance of ornithine increase? Vitamin B12 deficiency seems to induces changes in the expression of several enzymes at the transcriptional level (Arginase, Smd-1, Odc-1). Do the authors have any hypothesis regarding this issue?Author Response
Please see attachment.

Reviewer 3 Report
Comment:
1. Authors could add a scheme of derivatization reaction in the manuscript for amino acid and polyamine analysis.
2. Authors could add the chromatograms for amino acid and polyamine analyzed by HPLC.
3. Do authors detect the concentration of spermine? In polyamine cycle, spermine is important.
Round 2
Reviewer 1 Report
The authors have responded well to all of my queries and have presented a much stronger manuscript. I spotted a couple of typos detailed below, perhaps another proof read for grammatical and spelling errors would help.
Abstract:
Page 1, line 14: change metabolisms to metabolism
Introduction
Page 1, line 40: Change defect to defective
Reviewer 2 Report
The authors have answered satisfactorily to the questions raised by this reviewer.
Please check in lane 333, when you refer to “figure 5a and b” it should be “figure 6a and b”, if I am not wrong.
Reviewer 3 Report
The manuscript could be published in this form.